# Bacterial diversity and composition on the rinds of specific melon cultivars and hybrids from across different growing regions in the United States

Madison Goforth[1,2], Victoria Obergh[1,2], Richard Park[1,2], Martin Porchas[2,3], Kevin M. Crosby[2,4], John L. Jifon[2,4,5], Sadhana Ravishankar[1,2], Paul Brierley[2,3], Daniel L. Leskovar[2,4,6], Thomas A. Turini[2,7], Jonathan Schultheis[2,8], Timothy Coolong[2,9], Rhonda Miller[2,10], Hisashi Koiwa[2,4], Bhimanagouda S. Patil[2,4], Margarethe A. Cooper[1], Steven Huynh[11], Craig T. Parker[11], Wenjing Guan[2,5,12], Kerry K. Cooper[2,3,13] *

1 School of Animal and Comparative Biomedical Sciences, University of Arizona, Tucson, AZ, United States of America, 2 USDA National Center of Excellence for Melon at the Vegetable and Fruit Improvement Center of Texas A&M University, College Station, TX, United States of America, 3 Yuma Center of Excellence for Desert Agriculture, University of Arizona, Yuma, AZ, United States of America, 4 Vegetable & Fruit Improvement Center, Department of Horticultural Sciences, Texas A&M University, College Station, TX, United States of America, 5 Texas A&M AgriLife Research and Extension Center, Weslaco, TX, United States of America, 6 Texas A&M AgriLife Research and Extension Center, Texas A&M System, Uvalde, TX, United States of America, 7 University of California Cooperative Extension, Fresno, CA, United States of America, 8 Department of Horticultural Sciences, North Carolina State University, Raleigh, NC, United States of America, 9 Department of Horticulture, University of Georgia, Athens, GA, United States of America, 10 Department of Animal Science, Texas A&M University, College Station, TX, United States of America, 11 Produce Safety and Microbiology, Agricultural Research Service, USDA, Albany, CA, United States of America, 12 Southwest Purdue Agricultural Center, Vincennes, IN, United States of America, 13 BIO5 Institute, University of Arizona, Tucson, AZ, United States of America

* kcooper@arizona.edu

**Data Availability Statement:** All sequence files are available through the NCBI's Sequence Read Archive (SRA) under the Accession numbers:

## Abstract

The goal of this study was to characterize the bacterial diversity on different melon varieties grown in different regions of the US, and determine the influence that region, rind netting, and variety of melon has on the composition of the melon microbiome. Assessing the bacterial diversity of the microbiome on the melon rind can identify antagonistic and protagonistic bacteria for foodborne pathogens and spoilage organisms to improve melon safety, prolong shelf-life, and/or improve overall plant health. Bacterial community composition of melons (n = 603) grown in seven locations over a four-year period were used for 16S rRNA gene amplicon sequencing and analysis to identify bacterial diversity and constituents. Statistically significant differences in alpha diversity based on the rind netting and growing region (p < 0.01) were found among the melon samples. Principal Coordinate Analysis based on the Bray-Curtis dissimilarity distance matrix found that the melon bacterial communities clustered more by region rather than melon variety ($R^2$ value: 0.09 & $R^2$ value: 0.02 respectively). Taxonomic profiling among the growing regions found *Enterobacteriaceae*, *Bacillaceae*, *Microbacteriaceae*, and *Pseudomonadaceae* present on the different melon rinds at an abundance of $\geq$ 0.1%, but no specific core microbiome was found for netted melons. However, a core of *Pseudomonadaceae*, *Bacillaceae*, and *Exiguobacteraceae* were

SRR25178482, SRR25178481, SRR25178480 and associated with BioProject Accession number: PRJNA957757.

**Funding:** This study was supported by the USDA-NIFA-SCRI # 2017-51181-26834 through the National Center of Excellence for Melon at the Vegetable and Fruit Improvement Center of Texas A&M University. Technology and Research Initiative Fund (TRIF) provided to Kerry Cooper by the University of Arizona. No funding agency had any role in the study design, data collection and analysis, decision to publish, or preparation of the manuscript.

**Competing interests:** The authors have declared that no competing interests exist.

found for non-netted melons. The results of this study indicate that bacterial diversity is driven more by the region that the melons were grown in compared to rind netting or melon type. Establishing the foundation for regional differences could improve melon safety, shelf-life, and quality as well as the consumers' health.

## Introduction

Commercial melon production has risen in popularity as a commodity crop that provides essential nutrients like carbohydrates, water and vitamins [1, 2]. Production has risen to more complex varieties of melons that include hybrids and specialty melons that are modified to be low maintenance and suitable for warmer climates [3]. This shift in production regions that can suite the genetic changes in a melon allow for reducing chances in bacterial and fungal diseases like wilt and rot that can help increase production [3, 4]. In early 2000s, melons were the third most desired commodity crop in the US with the top three melons being cantaloupe, watermelon, and honeydew, which amounted to over $703.1 million dollars in revenue prior to 2006, however, in 2020 that number rose to $915 million dollars [1, 5]. Global production of melons continues to rise, from a 1.6% increase in 2018 to a total of 111 million tons [6]. For the United States, melon production increased from 1.95 million tons in 2018 to 2.7 million tons in 2020 [5–7]. In 2021, within the US, California and Arizona were the top two producers of cantaloupes, producing approximately 90% of all the US cantaloupes [5], with additional states like Georgia, Indiana, Florida, and Colorado making up the additional 10% [8, 9]. US cantaloupe production in these two regions alone had an economic value of $249.8 million dollars in 2021, with other farms making up the additional $27.8 million dollars [2, 5].

Prior to 2008, there have been 42 outbreaks caused by *Salmonella spp.* and two by *Listeria spp.* associated with the consumption of contaminated melons [10]. In the US, from 1971 to 2007 there was a multistate melon-associated outbreak almost every year, but since 2008 improved melon safety has decreased it to every two and half years [11]. However, even with improved melon safety there have been multistate outbreaks associated with melons including five *Salmonella* spp. outbreaks and the major 2011 *L. monocytogenes* outbreak that affected the whole melon industry [11–14]. Post-harvest methods include some critical control points for melon safety, including packaging and persistent cooling in between transportation and shipments to wholesale and retail stores [8, 13, 15]. Other factors that can contribute to melon contamination by *Salmonella spp.* and *Listeria spp.* include water quality, surface moisture, and sanitation of packaging handlers and facilities as well as the acidic pH of the melon rind, presence of inhibitor microbes, sugar content, and whether the melon is whole or pre-cut [1, 15–17]. Since 2019, there has not been a multistate outbreak related to melons, showing further strides made in pre-harvest and post-harvest methods. However, the consistent presence of these microorganisms and their threat on consumer health and growers costs should be continuously monitored [18].

To date, there is little known about the melon microbiome overall, but specifically only a couple of studies have looked at the bacterial communities present on the rinds, flesh, or stems of melons using non-culturing methods like next generation sequencing of 16S rRNA gene amplicons or shotgun metagenomics. Previous culture-based methods were successful at isolating bacteria that cause disease in melons [19, 20]. Although there are great methods in place for identifying microorganisms that are culturable, only 1% of bacteria are culturable, leaving about 99% microorganisms unidentified. Non-cultural methods are important for identifying

these bacteria and deciphering the melon microbiome. However, there are currently only three studies that have used next generation sequencing to explore the overall melon microbiome.

In 2021, Franco-Frías et al examined two farms in Coahuila, Mexico over the course of two months, collecting cantaloupe rind rinsates, workers' hand rinsates, and soil samples. The authors found that across field locations, there was no difference between rind, hands, and soil nor unique microbial species. Nevertheless, there was grouping among the cantaloupe and hand rinsates, while soil samples clustered by month sampled rather than the farm that the soil was taken from during the study. The study found all microbiome samples were dominated by the phyla *Proteobacteria*, *Firmicutes*, *Actinobacteria*, and *Bacteroidetes* [21]. In 2023, Xiao et al also conducted a 3-year study on the rhizospheres and endophytes of six varieties of oriental non-netted melons and netted melons that were all grown at the same location. The authors found that the bacterial diversity of the rhizospheres of the netted, oriental, and bulk soil samples were not different, and that these sample types were found clustered in the PCoA analysis individually by the type of sample rather than any other factor. However, there were unique dominant endophytic genera in netted melon rhizosphere samples that varied between netted and oriental stem samples [22]. Saminathan et al [23] characterized the bacterial communities of mature watermelon cultivars using 16S rRNA amplicon sequencing and metatranscriptomics from six varieties of watermelon grown at West Virginia State University over a two-season period. In the flesh of ripe watermelons, the dominant phyla were Proteobacteria with high expression of genes linked to carbohydrate metabolism of certain glucose and raffinose pathways [23]. These studies lay the foundation for characterizing bacterial diversity and composition on the rinds, flesh, and stems of melons.

It is important for the melon industry to understand what microbes are carried on the melon rind from pre-harvest to post-harvest from a food safety perspective as well as a spoilage aspect. Identifying the pre-harvest melon microbiome may give insights for important bacteria that could negatively impact colonization of *Listeria* and *Salmonella* as well as spoilage fungal and bacterial species. The goal of this study was to characterize bacterial diversity of specially bred melon varieties and hybrids grown in various locations across the United States. Specifically, we aimed to 1) determine the diversity and composition among netted and non-netted melons; 2) characterize growing regional differences in bacterial diversity and composition; and 3) determine bacterial compositional changes over a temporal shift among melons grown in the major producing regions of Arizona and California. Our study in conjunction with previous research lays the foundation for potential tools for melon growers, the melon industry, and the consumers by beginning to unravel what is on the surfaces of melons and how it is influenced by different environmental factors.

## Materials and methods

### Melon types

Over a four-year period from 2018 to 2021, samples were collected from 41 different melon varieties composed of typical commercial varieties and special hybrids generated for a large USDA funded study. These included 38 varieties of netted melons comprised of Alaniz Gold, Aphrodite, Athena, Caribbean King, Cruiser, Davinci, F39, Harper, Infinite Gold, Kiss SRK, Primo, S-Ac, S-Ma, S-Sa, S-Ta, S-Su, S-Sw, S-Tr, TH1, TH10, TH11, TH12, TH13, TH14, TH16, TH17, TH18, TH19, TH2, TH20, TH21, TH3, TH4, TH5, TH6, TH7, TH8, TH9, and 3 variety of non-netted melons comprised of OC164, HD150, and HD252. Melons were grown to the point of maturity in University Agricultural Experiment Station fields in seven different locations across the United States including California, Arizona,

Texas-Uvalde, Texas-Weslaco, Indiana, Georgia, and North Carolina (locations of the growing fields is demonstrated in S1 Fig) and were then shipped to the University of Arizona for processing as described below. All melons, including different types, from a particular location were planted, grown, and harvested in parallel in the same field under identical growing conditions. Table 1 lists the number of each variety from each location that was sampled for the study.

## Sample collection and DNA extraction

Upon arrival at the University of Arizona, a 5x5 cm$^2$ region of the rind was swabbed with a sterile swab soaked in sterile detergent solution (0.15 M NaCl, 0.1% Tween-20) [24, 25], which helped to dislodge bacteria from the surface of the melon rind. A limited area was taken for the melon microbiome due to the need of the melons for other experiments, but the size of the region was consistent across all melons sampled during the study. The collected swabs were then directly used for DNA extraction using the Qiagen DNeasy PowerSoil Pro kit (Qiagen, Hilden, Germany) per the manufacturer's instructions with the following modifications. (1) an additional step was added after solution C1 had been added, where samples were incubated at 65˚C for ten minutes; (2) to elute the DNA, 50 μL was added twice to the column filter for each sample with a centrifugation in between to increase yield.

## 16S rRNA gene PCR amplification and Illumina library preparation

The V4-V5 region of the 16S rRNA gene was PCR amplified in triplicate 25 μL reactions using barcoded 515F-926R primers. The 25 μL reaction was made up of 5 μL of template DNA, 1 μL of barcoded primers (10 μM), 1.25 μL of mPNA blocker (5 μM; mitochondria blockers; PNA Bio), 1.25 μL of pPNA blocker (5 μM; chloroplast blockers; PNA Bio), 6.5 μL of PCR grade nuclease-free water (Qiagen), and 10 μL of 2x Platinum Hot Start DNA Polymerase (Thermo Fisher Scientific, Waltham, MA). Each of the three PCR reactions were run at 95˚C for 3 minutes, followed by 30 cycles of 45s at 95˚C, 45s at 50˚C, 90s at 68˚C, and a final amplification step of 68˚C for 5 min on three separate thermocyclers. PCR grade water was utilized as negative controls for all reactions. Corresponding PCR products from the triplicate reactions for each sample were then pooled together and visualized on a 1.5% agarose gel for confirmation of proper amplification and negative controls were examined to confirm no contamination of the reactions. Samples were then quantified using the Quant-iT PicoGreen dsDNA assay (Invitrogen, Waltham, MA, USA) per the manufacturer's instructions. Individual barcoded sequencing libraries for each sample were pooled together in equal molar ratios and then cleaned using the QIAquick PCR cleanup kit (Qiagen, Hilden, Germany) per the manufacturer's protocol. Cleaned and pooled samples were then sequenced on an Illumina MiSeq with the reagent v3 kit (600 cycle) to generate 300 bp paired end reads for each sample. All sequence reads are available through the NCBI's Sequence Read Archive (SRA) under the Accession numbers: SRR25178482, SRR25178481, SRR25178480 and associated with BioProject Accession number: PRJNA957757.

## Sequencing read processing

All sequence reads were demultiplexed and quality trimmed to ≥Q30 for forward and reverse reads, then denoised after trimming, and finally merged using the DADA2 plugin [26] in QIIME2 software (v2020.2) [27]. Merged reads were used for further analysis of samples as described below.

**Table 1. Varieties and locations of melon samples collected during the study.**

| Melon variety | Arizona | California | Texas-Uvalde | Texas-Weslaco | North Carolina | Georgia | Indiana | TOTAL |
|---|---|---|---|---|---|---|---|---|
| Alaniz Gold | 3 | 0 | 0 | 0 | 0 | 0 | 0 | 3 |
| Aphrodite | 0 | 0 | 0 | 0 | 0 | 3 | 0 | 3 |
| Athena | 0 | 0 | 0 | 0 | 3 | 3 | 0 | 6 |
| Caribbean King | 0 | 3 | 0 | 0 | 0 | 0 | 0 | 3 |
| Cruiser | 0 | 0 | 0 | 3 | 0 | 0 | 0 | 3 |
| Davinci | 9 | 8 | 6 | 3 | 9 | 6 | 6 | 47 |
| F39 | 12 | 9 | 3 | 3 | 9 | 3 | 3 | 42 |
| Harper | 4 | 14 | 0 | 0 | 0 | 0 | 0 | 18 |
| HD150 | 6 | 3 | 6 | 3 | 3 | 3 | 3 | 27 |
| HD252 | 6 | 3 | 3 | 3 | 6 | 3 | 3 | 27 |
| Infinite Gold | 10 | 5 | 3 | 3 | 6 | 3 | 3 | 33 |
| Kiss SRK | 6 | 0 | 0 | 0 | 0 | 0 | 0 | 6 |
| Oc164 | 6 | 0 | 5 | 3 | 3 | 3 | 3 | 23 |
| Primo | 0 | 0 | 0 | 3 | 0 | 0 | 0 | 3 |
| S-Ac | 3 | 3 | 3 | 0 | 0 | 0 | 3 | 12 |
| S-Ma | 0 | 3 | 3 | 0 | 0 | 0 | 3 | 9 |
| S-Sa | 3 | 3 | 3 | 0 | 0 | 0 | 0 | 9 |
| S-Ta | 0 | 0 | 3 | 0 | 0 | 0 | 3 | 6 |
| S-Su | 3 | 3 | 0 | 0 | 0 | 0 | 3 | 9 |
| S-Sw | 3 | 3 | 0 | 0 | 0 | 0 | 3 | 9 |
| S-Tr | 0 | 3 | 3 | 0 | 0 | 0 | 3 | 9 |
| TH1 | 9 | 8 | 12 | 0 | 3 | 0 | 3 | 35 |
| TH10 | 6 | 6 | 9 | 0 | 3 | 0 | 3 | 27 |
| TH11 | 3 | 2 | 6 | 0 | 0 | 0 | 0 | 11 |
| TH12 | 9 | 4 | 3 | 0 | 3 | 0 | 0 | 19 |
| TH13 | 6 | 5 | 9 | 0 | 0 | 0 | 0 | 20 |
| TH14 | 3 | 0 | 6 | 0 | 0 | 0 | 0 | 9 |
| TH16 | 3 | 2 | 3 | 0 | 3 | 0 | 0 | 11 |
| TH17 | 3 | 3 | 0 | 0 | 3 | 0 | 0 | 9 |
| TH18 | 3 | 2 | 0 | 0 | 3 | 0 | 0 | 8 |
| TH19 | 3 | 3 | 0 | 0 | 3 | 0 | 0 | 9 |
| TH2 | 3 | 3 | 0 | 0 | 3 | 0 | 0 | 9 |
| TH20 | 3 | 0 | 0 | 0 | 4 | 0 | 0 | 7 |
| TH21 | 3 | 0 | 0 | 0 | 0 | 0 | 0 | 3 |
| TH3 | 3 | 3 | 0 | 0 | 3 | 0 | 0 | 9 |
| TH4 | 3 | 3 | 0 | 0 | 3 | 0 | 0 | 9 |
| TH5 | 9 | 9 | 9 | 0 | 6 | 0 | 3 | 36 |
| TH6 | 9 | 8 | 9 | 0 | 6 | 0 | 0 | 32 |
| TH7 | 3 | 0 | 0 | 0 | 0 | 0 | 0 | 3 |
| TH8 | 3 | 0 | 0 | 0 | 0 | 0 | 0 | 3 |
| TH9 | 6 | 6 | 9 | 0 | 3 | 0 | 3 | 27 |
| TOTAL | 167 | 130 | 116 | 24 | 88 | 27 | 51 | 603 |

## Taxonomic assignment

Taxonomic classification for sequence reads were done in QIIME2 with the feature-classifier plugin and Greengenes database (v13.8) with 99% sequence similarity. The classifier was trained using 515F primer 5′-GTGYCAGCMGCCGCGGTAA-3′ and 926R primer 5′-

CCGYCAATTYMTTTRAGTTT-3' with a minimum length of 200 bp and maximum length of 500 bp.

## Alpha and beta diversity analysis

The amplicon sequence variant (ASV) file and phylogenetic tree file generated in QIIME2 were exported and then imported into R (v4.2.1) along with the metadata file for further analysis using the following packages: (1) phyloseq (v.1.40.0) [28], (2) microbiome (v.1.18.0) [29], and (3) vegan (v.2.6.2) [30]. Initially, all samples were filtered for chloroplasts and mitochondria to remove any of those sequences not blocked during the amplification step with the PNA blockers, after filtering out these sequences all the samples were rarefied to 1,500 sequencing depth that removed 102/603 samples from further analysis. The samples that were removed had also included negative control PCR water blanks. These water blanks were also used as a base for any contamination of reads that were removed from all samples. After rarefying, the alpha diversity and beta diversity were determined using phyloseq and microbiome packages. The Shannon index with Kruskal-Wallis and Wilcox statistical tests to compare alpha diversity was done using the phyloseq and vegan packages. Beta diversity was determined using the Bray-Curtis dissimilarity distance matrix using the phyloseq and microbiome packages, and variability among samples for bacterial composition was tested using PERMANOVA with adonis2 with 999 permutations from the vegan package.

## Taxonomic composition and visualization, core microbiome, and Linear Discriminant Analysis (LDA) Effect Size (LEfSe) analysis

Taxonomic composition analysis was determined using the phyloseq, microbiome, microbiomeutilities [28, 31], and vegan packages and then visualized using different categorical variables at 0.1% relative abundance and 75% prevalence for parameters. The core microbiome was determined using the parameters of 0.001 detection and 75% prevalence using the phyloseq package, and was calculated for all melon samples, only netted melons, only non-netted melons, and the growing region of the melons. Linear Discriminant Analysis Effect Size (LEfSe) was used to determine top taxonomic features to assess differences between samples, which was calculated for the melon samples using the lefser (v.1.16.0) [32] package in addition to the microbiomeMarker (v.1.3.2) [32, 33] with a LDA cutoff at four (4).

## Temporal analysis

Samples from Arizona and California were temporally analyzed because there was consistent sampling for netted melons across all four years. Dissimilarity matrices for Arizona and California netted melons bacterial communities were computed using beta-dispersion with the vegan and betapart (v.1.5.6) [34] packages for Simpson, Sorensen, SNE, and Bray-Curtis Dissimilarity. Beta-dispersion was tested against Euclidean distances (the four years that netted melons were collected) against the dissimilarity matrices above to look for the presence or absence of species (Simpson), replacement/turnover of species (Sorensen), nestedness of species (SNE), and composition variance of species (Bray-Curtis Dissimilarity). Additionally, from Simpson and SNE calculations, β-total dissimilarity was also calculated. Mantel distance correlations were done with the vegan package using the dissimilarity matrices produced from the beta-dispersion against Euclidean temporal distances and plotted for variation in bacterial dissimilarities of netted melons among Arizona and California.

## Results

### Non-netted and netted melons

First, we looked at bacterial diversity differences based on the melon rind netting to determine the influence the rind composition might have on the bacterial communities. A comparison of the Shannon diversity evenness (alpha diversity) between the two types of melon rinds found that there was a statistically significant difference between the two rind types, with higher diversity found on melons with non-netted rinds (Fig 1A; p-value: <0.01). However, there were also differences based on the year the samples were collected, where there were more differences in bacterial diversity among netted melons sampled during the years 2018, 2019, and 2020 compared to netted melons of 2021 (p-value < 0.01). There were only significant bacterial composition differences among non-netted melons from 2018 to non-netted melons from 2019, as well as non-netted melons from 2018 to netted melons from 2021 (Fig 1B; p-value <0.01). Non-netted melon samples were not collected during 2020 and 2021. Beta diversity analysis demonstrated some clustering based on the type of melon rind, but overall dispersal showed bacterial community composition is relatively similar among the two types of rind netting (S2A Fig; $R^2$ value: 0.02; PERMANOVA p-value: 0.05; Permutations 999). The beta diversity supported the alpha diversity findings in that samples were found clustered by the year that the melons were sampled. (S2B Fig; $R^2$ value: 0.09; Permanova p-value 0.001; Permutations 999).

Among the rind netting, there was a greater richness of bacterial families for non-netted melons with a mean of 122 observed species, whereas observed species for netted melons was 80. Next, we explored the specific taxonomic features for the netted rind melons versus non-netted rind melons, and again this analysis demonstrated the variation in diversity between the two categories of melon rinds even when samples were split by location, which consisted of 16 families representing non-netted melons compared to four for netted melons (Fig 2A and 2B). Although all four bacterial families on netted melons were present on non-netted melons, but typically at different abundance levels (S2A Fig). Netted melons showed that in Indiana, North Carolina, Texas-Uvalde, and Texas-Weslaco there was a higher abundance of *Pseudomonadaceae* (Fig 2A). Indiana and Georgia had more relative abundance of *Microbacteriaceae* than other regions but a decrease in *Enterobacteriaceae* and *Bacillaceae*, which were both at higher relative abundance in regions like Arizona, California, North Carolina, and Texas-Uvalde. Non-netted melons grown in both Texas-Weslaco and Texas-Uvalde had more *Exiguobacteraceae* than other regions but a decrease in *Bacillaceae* compared to netted melons

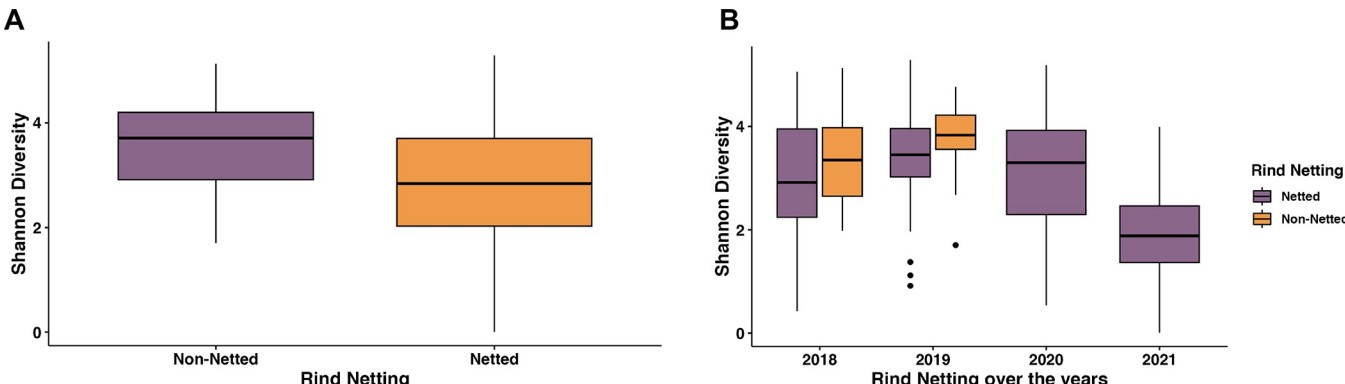

**Fig 1. Bacterial diversity and composition of the melon carposphere split by netting.** (A) Shannon diversity index plotted for the rind netting. (B) Shannon diversity index plotted based on the year for the rind netting.

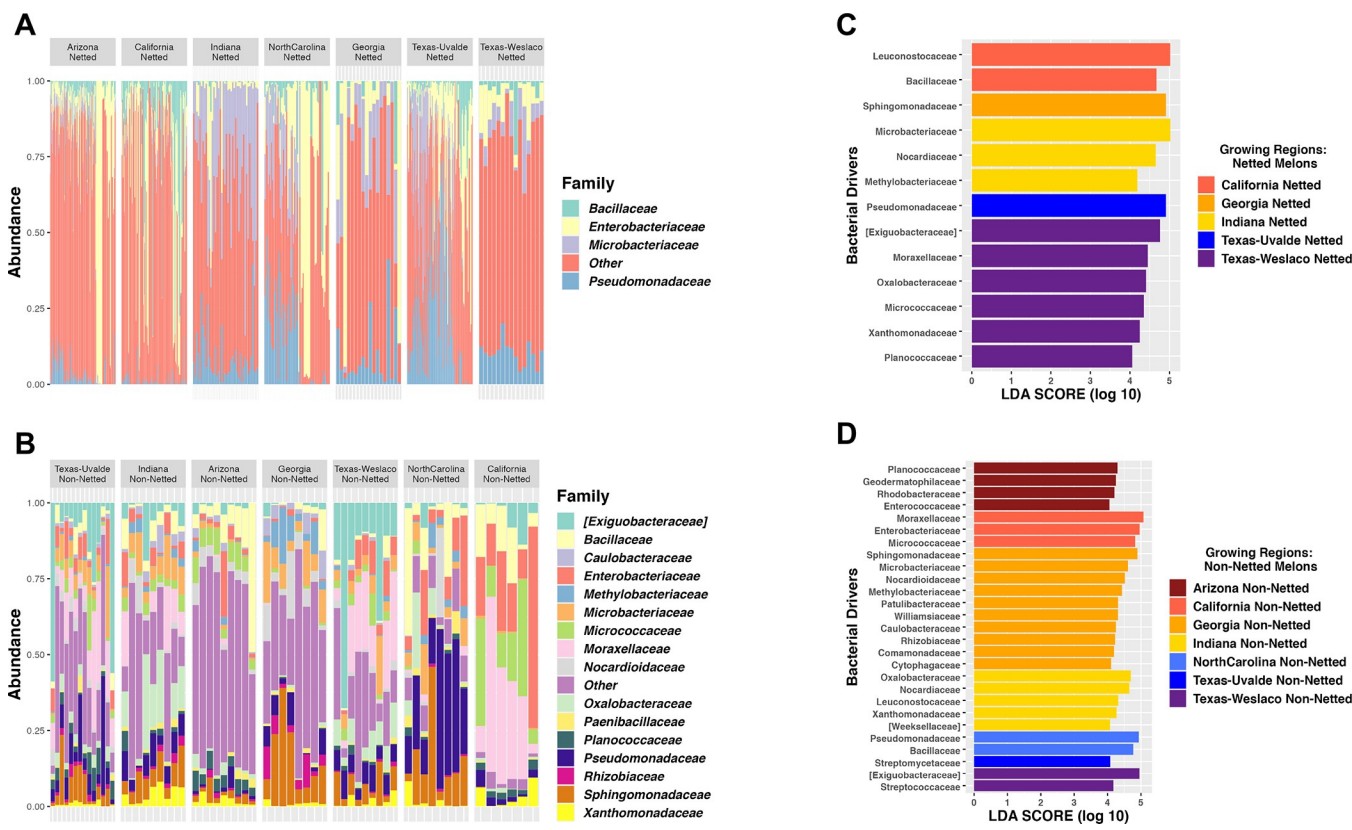

**Fig 2. Taxonomic profiling of netted and non-netted melons.** (A) Taxonomic relative abundance looking at the regions for netted melons. (B) Taxonomic relative abundance looking at the regions for non-netted melons. (C) LEfSe analysis for netted melons looking at sites (LDA = 4). (D) LEfSe analysis for non-netted melons looking at sites (LDA = 4).

(Fig 2B). Non-netted melons grown in Indiana had greater relative abundance among the different location samples of *Oxalobacteraceae*, whereas North Carolina grown netted and non-netted melons had similar relative abundance of *Pseudomonadaceae*. California non-netted melons and netted melons had a high relative abundance for *Enterobacteriaceae* compared to other bacterial families that were shared within the region. Arizona and Georgia were both found to have a large number of bacterial families that fell into the "Other" category, suggesting numerous bacterial families that are present just not at the abundance cut-off for the top bacterial families. Additionally, Georgia had a high relative abundance of *Sphingomonadaceae* and *Methylobacteriaceae* compared to melons grown in other regions, whereas Arizona had a high abundance of *Bacillaceae* and *Micrococcaceae* (Fig 2B).

LEfSe analysis further supported the general lower bacterial diversity on netted melons compared to non-netted melons as it identified only five regions for netted melons to have taxonomic features above an LDA score of 4, suggesting that there were few taxonomic features explaining the diversity differences for netted melons from different locations. Although some taxonomic features were identified between the locations, such as Indiana netted melons had *Microbacteriaceae* as the highest taxonomic feature, Georgia and Texas-Uvalde only had one taxonomic feature which was *Sphingomonadaceae* and *Pseudomonadaceae*, respectively, Arizona and North Carolina did not have any identified features, California netted melons had *Leuconostocaceae* as the top taxonomic feature, and Texas-Weslaco had *Exiguobacteraceae* that was consistent with the taxonomic relative abundance plot (Fig 2C). LEfSe analysis of non-

netted melons found all seven regions had taxonomic features that were responsible for the differences between locations. Taxonomic features for each region were consistent with the taxonomic relative abundance plot for the regions where the bacterial families were present in both analyses. Only Georgia, Indiana, and Texas-Weslaco had high taxonomic features that were consistent with the netted samples taken from these regions, including *Sphingomonadaceae*, *Nocardioidaceae*, and *Exiguobacteraceae*, respectively. Arizona had *Planococcaceae* as the top taxonomic feature, while California had *Moraxellaceae*, North Carolina had *Pseudomonadaceae*, and Texas-Uvalde had only *Streptomycetaceae* (Fig 2D). Core microbiome analysis at a relative abundance of 0.1% found no core regardless of rind netting. However, when splitting them by the rind netting, non-netted melons had three core taxa of *Pseudomonadaceae*, *Bacillaceae*, *Exiguobacteraceae*, while netted melons did not have any core taxa even at extremely low relative abundance ($1 \times 10^{-10}$) percentage cutoffs (S1 Table).

## Locations

As there appeared to be some significant differences in the diversity and taxonomic composition of the bacteria on melons grown in different locations regardless of the type of melon rind, we looked further into the impact location has on the bacterial diversity of the melon rind. The Shannon diversity evenness index indicated significant differences in the diversity among the regions regardless of the rind netting type, but also within the specific rind netting collected in each region compared to other regions. Texas-Uvalde had the greatest significance in bacterial diversity between netted and non-netted melons with Indiana following second (P-values: $2.4 \times 10^{-5}$ and $6 \times 10^{-4}$, respectively). The greatest difference in bacterial diversity was between one region's netted melons compared to another region's non-netted melons that always resulted in a significant difference (Fig 3A; all p-values < 0.01). PCoA based on the Bray-Curtis dissimilarity distance matrix provided a clearer distinction of growing regions having an impact on the composition of the bacterial community, as the samples from most regions clustered together. Yet, there was still overlap between the different regions indicating some similarity in composition across the regions. There were some interesting spatial distributions of the samples, as the western samples (Arizona and California) clustered closer together but separately from most central and eastern region samples. However, the samples with the widest dispersal were Texas-Uvalde and North Carolina, which clustered with both

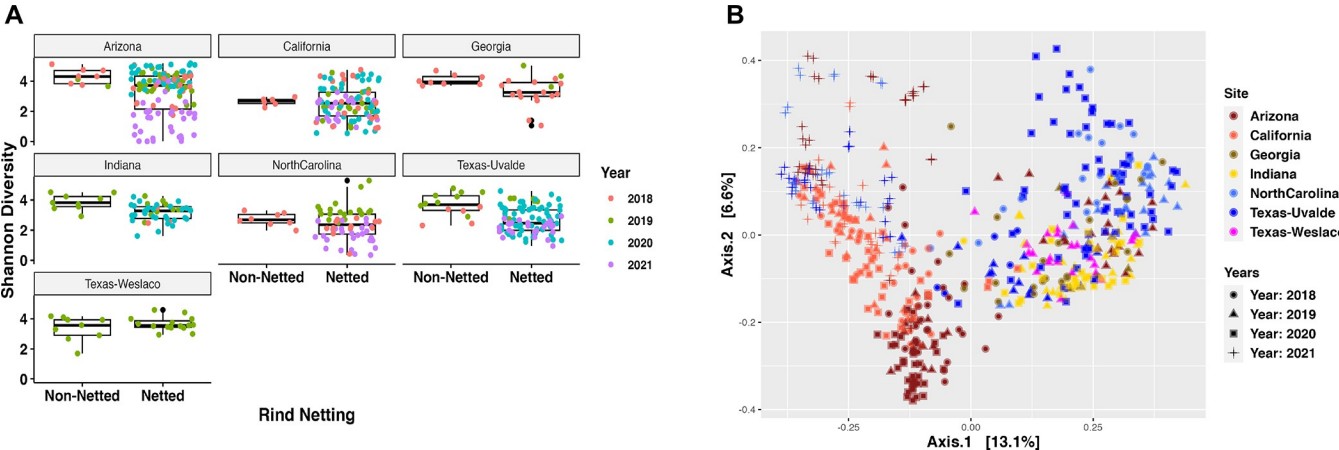

**Fig 3. Bacterial diversity of regional differences of the melon carposphere.** (A) Shannon diversity index plotted based on the site for the rind netting. (B) Bray-Curtis PCoA plot clustered by the region.

the Western and Eastern samples (Fig 3B; $R^2$ value: 0.09; PERMANOVA P-value: 0.001; Permutations 999).

Based on a time series analysis of the four years that samples were collected, Bray-Curtis PCoA dissimilarity plots of bacterial community composition by the years sampled show regions clustering for all years except for the year 2021. However, samples were limited for certain years particularly for 2020 and 2021, so further studies that can account for more samples from the different regions over several years could be more effective in considering a temporal difference in bacterial community clustering. The year 2019 had the biggest role in bacterial clustering ($R^2$ value 0.25; Permanova p-value 0.001; Permutations 999), whereas the year 2021 had the smallest role in bacterial community clustering ($R^2$ value 0.14; Permanova p-value 0.001; Permutations 999) but was limited by sample number. Overall, the year the samples were collected were found to have a significant impact in bacterial clustering compared to other variables (S3 Fig; $R^2$ value: 0.09; Permanova p-value 0.001; Permutations 999).

**Arizona.** Both netted and non-netted melons grown in Arizona had some of the highest levels of bacterial diversity according to the Shannon diversity index, with the netted being significantly lower than the non-netted samples (Fig 3A; p-value = 0.09). The taxonomic relative abundance plot for Arizona samples supported this diversity as both netted and non-netted samples had large relative abundance of "Other" thus indicating the communities were composed larger amounts of less dominant bacterial families (Fig 4A). However, the plots also demonstrated that four bacterial families with higher relative abundance, including *Bacillaceae*, *Enterobacteriaceae*, *Microbacteriaceae*, and *Pseudomonadaceae*, which were shared among both netted and non-netted samples but at different rates. For example, netted samples had more *Enterobacteriaceae* and *Pseudomonadaceae* whereas non-netted samples had a

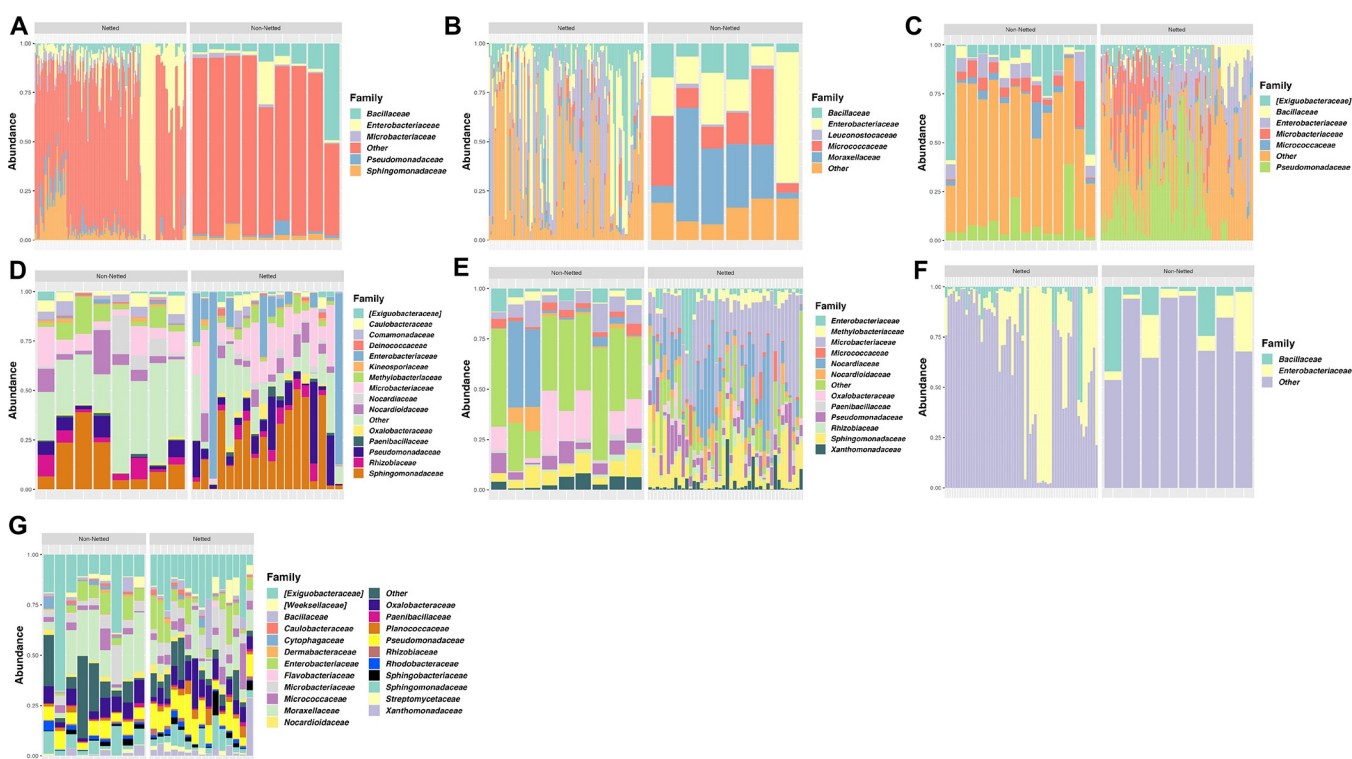

**Fig 4. Taxonomic profiling of the top bacterial families.** Families determined at 75% prevalence and 0.1% relative abundance for netted and non-netted melon carposphere at seven locations: (A) Arizona, (B) California, (C) Texas-Uvalde, (D) Georgia, (E) Indiana, (F) North Carolina, and (G) Texas-Weslaco.

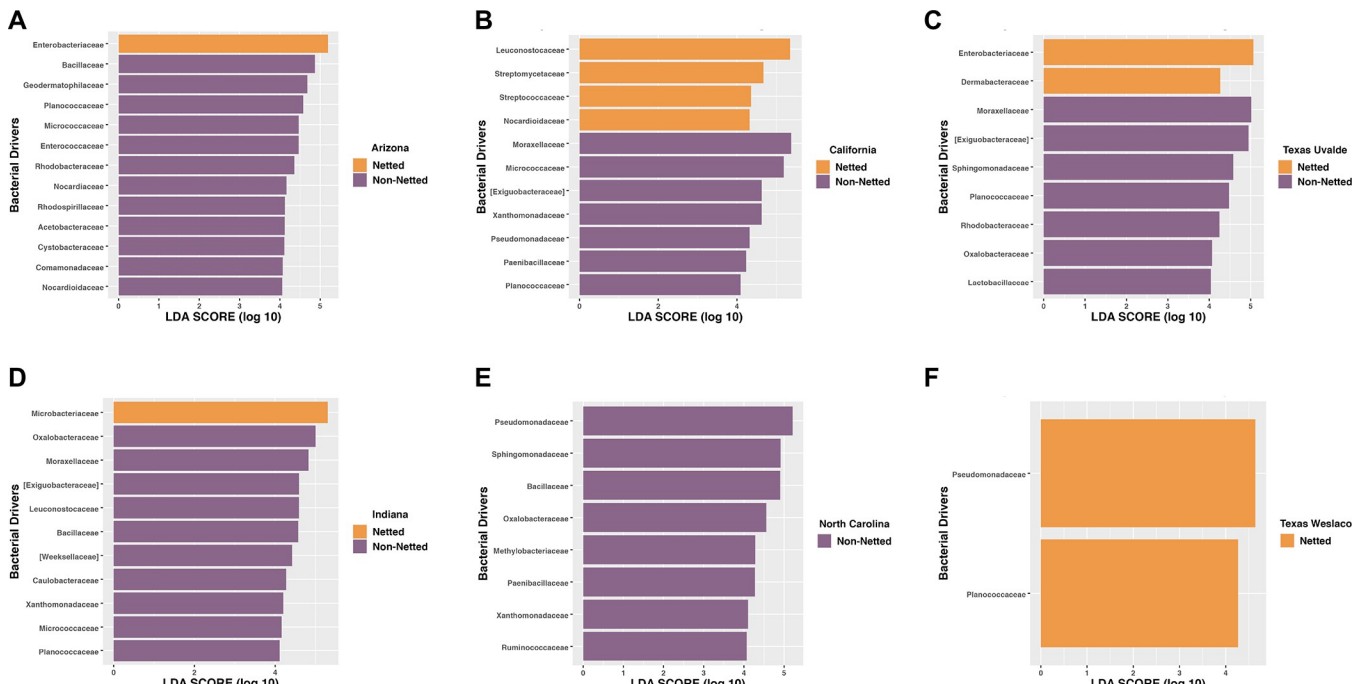

**Fig 5. Top taxonomic features of bacterial families for netted and non-netted melon carposphere at six locations.** (LDA score = 4): (A) Arizona, (B) California, (C), Indiana (D) Texas-Uvalde, (E) North Carolina, (F) Texas-Weslaco. Georgia was excluded based on the LDA cutoff.

decrease in these families (Fig 4A). The taxonomic features for both rind types were similar to the relative abundances of bacterial families, where the only feature for netted melons was *Enterobacteriaceae* (Fig 5A). Non-netted melons had 13 taxonomic features with *Bacillaceae* being the top feature and the rest possibly coming from the "Other" category within the samples from the taxonomic relative abundance plot (Fig 5A; LDA = 4). The core microbiome analysis supported the alpha diversity and relative abundance results in that no shared core taxa were found among the netted melons (S2 Table) but there were 13 core taxa at an abundance of 0.1% for the non-netted melon samples with the main taxonomic features appearing in the core and LEfSe analysis with a few exceptions (Fig 5A; S3 Table).

**California.** Both netted and non-netted melons grown in California had some of the lowest levels of bacterial diversity according to the Shannon diversity index, with the netted being lower than the non-netted samples, but no significance in bacterial diversity among the rind netting in this region (Fig 3A; p-value = 0.14). The taxonomic relative abundance plot showed five bacterial families that include *Leuconostocaceae* and *Moraxellaceae*, and netted melons had more relative abundance of *Leuconostocaceae*, while non-netted melons had more *Moraxellaceae* (Fig 4B). These were also consistent with being the top taxonomic features for both rind types in the LEfSe analysis (Fig 5B; LDA = 4). The number of taxonomic features varied as netted melons had four and non-netted melons had seven, and the only bacterial family that was present in both the relative abundance analysis and the LefSe for the non-netted melons was *Micrococcaceae* and *Leuconostocaceae* of non-netted melons (Fig 5B). Like the previous analysis, the core microbiome for both netted and non-netted melons was composed of just *Leuconostocaceae* and *Bacillaceae* (S2 Table). These same two bacterial families composed the entire core taxa for netted melons, but for non-netted melons the core taxa included an additional seven core taxa (S3 Table).

**Texas-Uvalde.** Both netted and non-netted melons grown in Texas Uvalde had the highest levels of bacterial diversity from the seven sampling locations according to the Shannon diversity index, with the netted melons being lower than non-netted resulting in high significance in bacterial diversity based on rind netting (Fig 3A; p-value = 2.4 x $10^{-5}$). The taxonomic relative abundance plot for netted melons showed six bacterial families that were similar with melons from California and Arizona, but *Enterobacteriaceae* was present at a higher relative abundance (Fig 4A–4C). Non-netted samples were quite consistent in relative abundance of bacterial families across the different samples compared to the netted melons, as some netted melons had higher abundances of *Microbacteriaceae* and/or a decrease in *Enterobacteriaceae* opposite was observed in the other half of the samples (Fig 4C). LefSe analysis showed that netted melons had two taxonomic features identified as *Enterobacteriaceae* and *Dermabacteraceae*, whereas non-netted melons had seven taxonomic features identified by LefSe analysis including the top two features as *Moraxellaceae* and *Exiguobacteraceae* with the later also being present on the relative abundance plot (Fig 5C). Overall melons grown in Texas-Uvalde had a single core taxon of *Exiguobacteraceae*, whereas netted melons also had a single core taxon of *Enterobacteriaceae*. However, non-netted melons shared a core microbiome of nine bacterial families including both *Exiguobacteraceae* and *Enterobacteriaceae* (Fig 5C, S2 and S3 Tables).

**Georgia.** Both netted and non-netted melons grown in Georgia had been the third highest for Shannon diversity index levels among the different locations with samples close to levels observed for Arizona grown melons, but with a smaller spread based on netting type. Like other locations, the netted melons had a lower Shannon diversity index than the non-netted samples with a low significant difference based on the rind netting (Fig 3A; p-value = 0.07). The taxonomic relative abundance plots found 15 bacterial families, 12 of which were identified in other locations, and 3 new families like *Comamonadaceae*, *Deinococcaceae*, and *Kineosporiaceae*. *Sphingomonadaceae* and *Microbacteriaceae* were present at high relative abundances on both types of melon netting, whereas *Enterobacteriaceae* was present at higher abundance only on netted melons and *Caulobacteraceae* only on non-netted melons (Fig 4D). LefSe analysis could not identify any taxonomic features for Georgia even with smaller cutoff values. The overall core taxa for melons grown in Georgia regardless of melon netting type contained seven bacterial families, and non-netted melons had five of these bacterial families plus *Geodermatophilaceae* and *Bacillaceae* and netted melons included five families plus *Enterobacteriaceae* (S2 and S3 Tables).

**Indiana.** Both netted and non-netted melons grown in Indiana had Shannon diversity index levels similar to that of Georgia, but with a significant difference between the netted versus non-netted samples (Fig 3A; p-value = 6x$10^{-4}$). Taxonomic analysis found 12 bacterial families that were also identified in Georgia and Texas-Weslaco grown melons like *Xanthomonadaceae* and *Sphingomonadaceae* (Fig 4E–4G). Netted melons grown in Indiana were found to have a higher relative abundance of *Microbacteriaceae* and some melons also having higher levels of *Nocardiaceae*, whereas non-netted melons had a decrease in relative abundance of these specific families that was often replaced with an increase in the general "Other" category and/or *Oxalobacteraceae* (Fig 4E). LefSe analysis found the top taxonomic features were *Microbacteriaceae* and *Oxalobacteraceae* for netted and non-netted melons, respectively, and overall non-netted melons had ten additional taxonomic features compared to the single taxonomic feature for the netted melons (Fig 5D; LDA = 4). Core analysis identified seven bacterial families shared among netted and non-netted melons grown in Indiana, whereas 16 core bacterial families were present in non-netted melons and eight in netted melons (S2 and S3 Tables).

**North Carolina.** Netted and non-netted melons grown in North Carolina had some of the lowest levels of bacterial diversity based on the Shannon diversity index, which were

similar to diversity levels observed in melons grown in California, with netted melons being lower than the non-netted but no overall significance based on type of rind netting (Fig 3A; p-value = 0.31). The taxonomic relative abundance plot had only two families, *Bacillaceae* and *Enterobacteriaceae* (Fig 4F). Among the netted melons, 63% of the samples were made up of mostly "Other" and a lower level of *Enterobacteriaceae*, while 37% of the samples had high levels of *Enterobacteriaceae*. In non-netted melons most of the relative abundance was made up of the "Other" category with various levels of *Enterobacteriaceae* and *Bacillaceae*. The LefSe analysis found eight taxonomic features but only for non-netted melons, with the top feature being *Pseudomonadaceae* followed by *Sphingomonadaceae* (Fig 5E). The core analysis found only *Bacillaceae* shared among the netted and non-netted melons. There was one core taxon for netted melons which was *Bacillaceae*, whereas there were six taxa for non-netted melons including four of the taxonomic features (S2 and S3 Tables).

**Texas-Weslaco.** Both netted and non-netted melons grown in Texas-Weslaco had high levels of bacterial diversity compared to other sampling regions based on the Shannon diversity index, and no significant difference was observed in bacterial diversity based on the rind netting (Fig 3A; p-value = 0.91). The taxonomic relative abundance plot showed 22 bacterial families that were present in both the netted and non-netted melons grown in Texas-Weslaco. Both rind types were similar for relative abundances of bacteria with the highest relative abundance coming from *Exiguobacteraceae*, while some samples did have higher levels of *Oxalobacteraceae* and *Pseudomonadaceae* (Fig 4G). Compared to the other locations there was a smaller relative abundance in *Enterobacteriaceae* among all the different samples, although it was still present, whereas only netted melons had taxonomic features that included *Pseudomonadaceae* and *Planococcaceae* (Fig 5F). Core taxa analysis found 15 bacterial families that were shared among both netted and non-netted melons grown in Texas—Weslaco, which increased to 16 bacterial families when only examining non-netted melons (Fig 4G and S2 and S3 Tables).

**Arizona and California.** As these two locations produce approximately 90% of the melons for commercial use, we had the largest set of samples from these locations across the four-year study period, thus we decided to analysis these samples further. Shannon evenness index showed there was a significant difference in bacterial diversity among Arizona netted and California netted melons (p-value < 0.001). There were also significant differences in diversity among Arizona non-netted melons and California netted and non-netted melons (Fig 6A; all p-values < 0.01). PCoA analysis based on Bray-Curtis dissimilarity mainly grouped samples

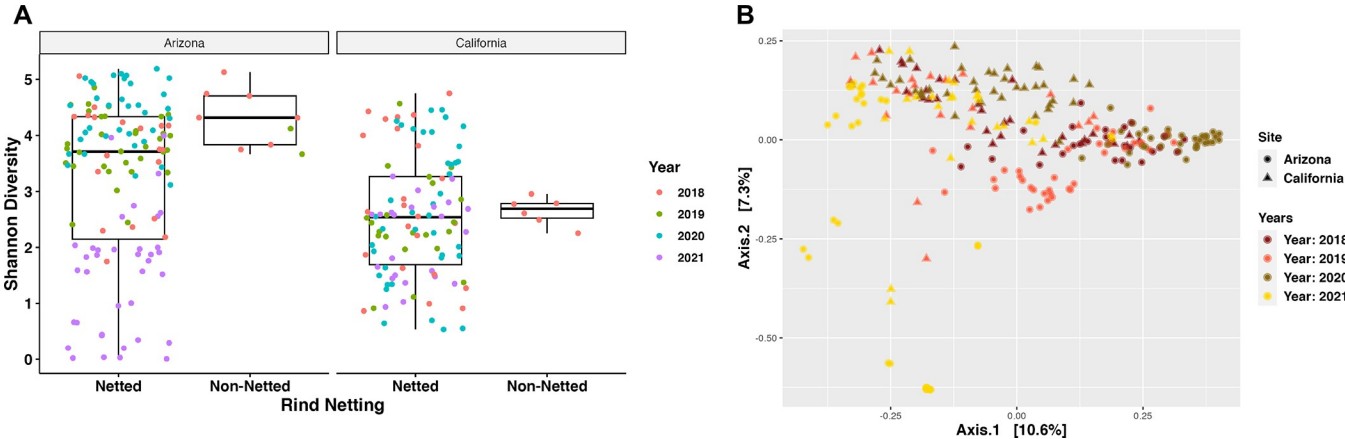

**Fig 6. Bacterial diversity and composition of the melon carposphere based on two commercial agricultural states.** (A) Shannon diversity index plotted based on Arizona and California for the rind netting. (B) Bray-Curtis PCoA plot clustered by the regions, Arizona, and California.

by region with type of netting having little significance on bacterial composition differences (Site $R^2$: 0.04; Permanova p-value: 0.001; Netting $R^2$: 0.007; Permutations 999). Year to year sampling had a bigger role in bacterial community composition, possibly indicating temporal correlation with the bacteria present on the surfaces of melons (Fig 6B; Year $R^2$: 0.12; Permanova p-value: 0.001; Permutations 999).

Taxonomic relative abundance among netted and non-netted melons shared *Bacillaceae* and *Enterobacteriaceae*. For netted melons from both locations, *Enterobacteriaceae* had the highest relative abundance, but interestingly because of the high diversity on melons grown in Arizona, "Other" bacterial families dominated many of the netted melons particularly compared to California grown melons (S4A Fig). In non-netted samples, there were nine bacterial families or six more than netted melons, but the bacterial families found on netted melons were also present on non-netted melons, except for *Leuconostocaceae*. Comparing the taxonomic composition of the non-netted melon rinds between the regions found that Arizona grown melons had more *Planococcaceae*, while California grown melons had more *Micrococcaceae* (S4B Fig). Furthermore, Arizona grown melons had more taxonomic features identified by LEfSe analysis compared to California grown melons for both rind netting types. The top three taxonomic features for Arizona grown netted melons were *Sphingomonadaceae*, *Oxalobacteraceae*, and *Pseudomonadaceae*, while the top three for non-netted melons included *Geodermatophilaceae*, *Nocardioidaceae*, and *Rhodobacteraceae*. The top three taxonomic features of California grown netted melons were different from Arizona, had higher LDA scores, and included *Leuconostocaceae*, *Bacillaceae*, and *Moraxellaceae*, whereas non-netted melons included *Moraxellaceae*, *Enterobacteriaceae*, and *Micrococcaceae* (S4C and S4D Fig). Core microbiome analysis among the two growing regions for the netted melons had no core taxa, while non-netted melons had four core taxa, and overall, there was no core shared among the two regions for both rind netting (S4 Table).

Finally, we determined if the seasonal changes in the bacterial diversity and composition for netted melons in the two major growing regions were due to presence and absence of bacteria, bacterial turnover, or nestedness using the Mantel correlation. Due to sample numbers, we were not able to conduct the analysis on the non-netted melon samples for the two locations. The analysis for California grown netted melons over the four-year period showed that Sorensen had the most influence on bacterial composition, meaning that there was a huge taxa difference in presence and absence (S5A Fig; 0.31). Arizona (0.41) had more correlation with Sorensen for Euclidean temporal distance over the four years compared to California (S5B Fig). Turnover (Simpson diversity) was the second most influential on bacterial composition (0.22) for California (S5C Fig), whereas there was no significance for Arizona (S5D Fig; 0.01; p-value: 0.23). However, the second largest dissimilarity index was SNE dissimilarity for Arizona, which meant that taxa nestedness was more apparent for Arizona's netted melons (S5F Fig; 0.36), more so than California (S5E Fig; 0.09). In the Beta-total dissimilarity plot, seasonal changes in the bacterial composition of samples for Arizona grown netted melons were at least 54% due to bacterial turnover, while California had 58% due to bacterial nestedness (Fig 7).

## Discussion

Previous examination of bacterial communities on melons identified increased bacterial diversity on melons with netted rinds as compared to non-netted rinds [22], however our study found the opposite with non-netted melons consistently having higher alpha diversity compared to netted melons. Although we found that bacterial diversity of non-netted melon rind was higher than netted rinds in this study, such differences could be the result of major differences in melon cultivars examined between the studies, the multiple locations within this

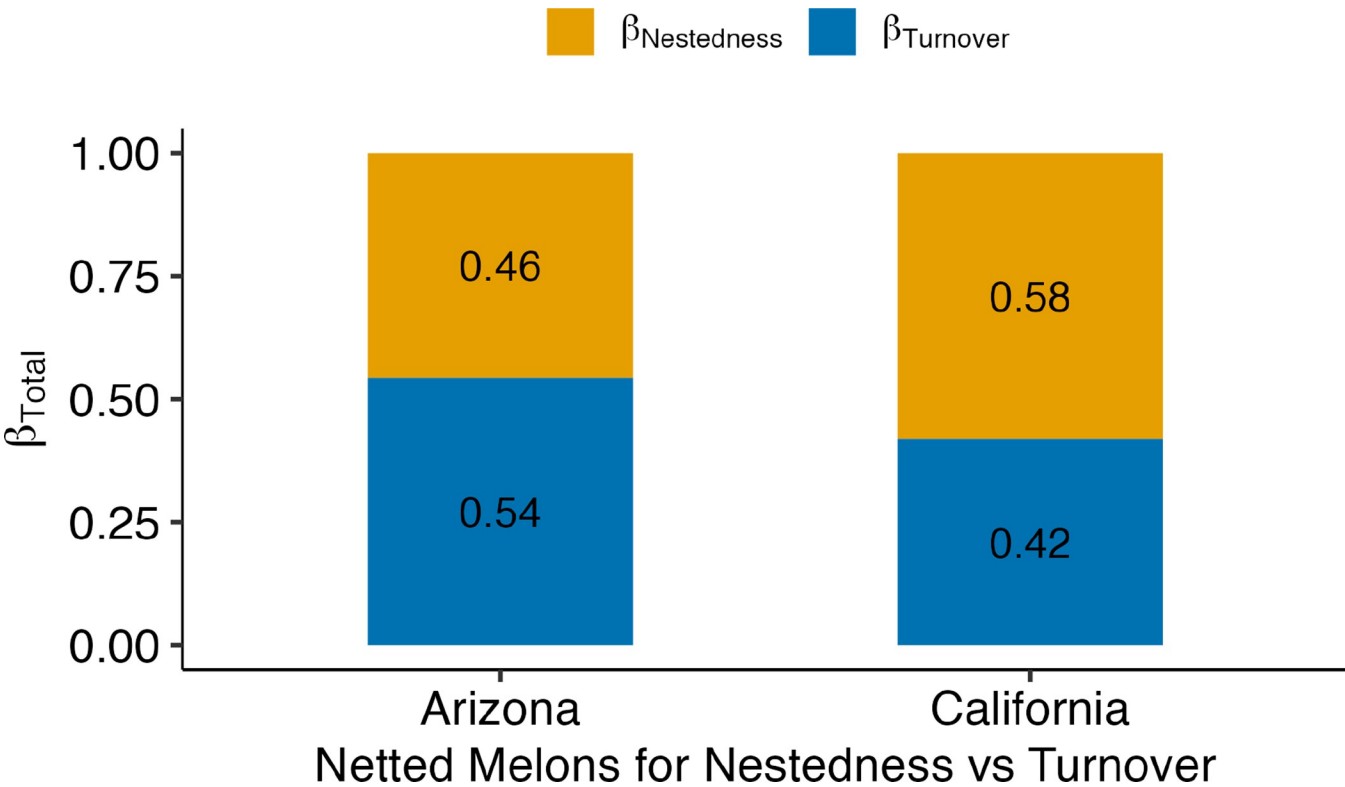

**Fig 7. Assessing species turnover (β-Simpson) or species nestedness (β- SNE) against total temporal distance.** Compositional variance (β-Total) for contribution of bacterial communities on Arizona and California netted melon surfaces. All years combined that the melons were collected during the study.

study, or our examination over several growth seasons. Although our results do agree with the results from Xiao et al, who also found that there was a difference among the communities based on the netting of the melons rather than the varieties [22], while we did not look specific at the impact that all the different cultivars in this study had on the bacterial diversity we did find that netting of the melons did play a role in the bacterial composition although not a major one.

In this study, we found that the bacterial diversity was impacted most by spatial variation and seasonal differences which seemed to impact bacterial composition significantly more than type of melon rind netting. In fact, the bacterial diversity patterns were most seasonal, as the year that the melons were harvested had stronger clustering patterns compared to rind netting or even growing region. Pre-harvest seasonal variations could be due to numerous factors such as changes in farming methods used between seasons, as well natural climate variation between seasons, and environmental factors (such as temperature) that could be site-specific and differ from season to season depending on location [35, 36].

Among all regions, both netted and non-netted melons shared *Bacillaceae*, *Enterobacteriaceae*, *Microbacteriaceae*, and *Pseudomonadaceae* at different relative abundances, but when further characterized by specific rind netting, only non-netted melons had a core for all regions, whereas netted melons did not have a core. LefSe analysis found *Enterobacteriaceae* as the top taxonomic feature for netted melons, while *Moraxellaceae* was the top feature for non-netted melons. Both bacterial families are Gram-negative and members of the phylum Pseudomonadota, renamed from Proteobacteria. Interestingly, Xiao et al found Proteobacteria to be the highest in both netted and oriental non-netted melons, similar to our findings [22].

Franco-Frías et al also found Proteobacteria to be the highest phylum present on cantaloupe rinds with Firmicutes being the second highest and lower levels of Actinobacteria [21]. However, our study found netted and non-netted melons had higher levels of Actinobacteria than Firmicutes, which could possibly be due to the standardization of land applications and/or fertilizers used in the different locations [9, 37].

In our study, bacterial communities clustered more by the region rather than the melon netting within the Bray-Curtis PCoA plot, indicating the critical spatial role has on bacterial diversity of melon rinds. Statistically, North Carolina had the most significant pairings for both non-netted melons and netted melons, and being an eastern coastal state there is potentially a different environment that is possibly exhibiting niche clustering compared to the other states like the southeastern coastal state of Georgia [38]. For example, certain climates like the arid desert in Arizona, or the relatively high humidity on the east coast could have a major role in how this clustering is being affected [39]. We saw did observe tighter clustering for Arizona and California while there was more of a spread for North Carolina and Texas-Uvalde over the four years of the study, but additional studies directly examining regional variation in rainfall or temperature are needed to draw conclusions about the impact on the melon microbiome. One study has found that differences that contributed to the shifts in bacterial diversity were greatly impacted by the region or location of the samples, indicating the environmental shifts due to the geographical location were influencing the diversity [38]. One study found specific factors like rain with dust carried fungal communities across regions that affected microorganisms adhering to the surfaces of produce [40]. Besides the intra-regional role in microbial transference, inter-regional factors like soil health and pH can contribute to bacterial composition impacting those communities that persist in an environment or host [21, 22]. However, additional research must be done to confirm the impact of each specific environmental factor has on the bacterial diversity of the melon microbiome.

Bacterial composition among all regions, except Arizona and Georgia, identified the bacterial family *Enterobacteriaceae* as a core taxon for non-netted melons, and was also a core taxon for netted melons among four regions, excluding Arizona, California, and North Carolina. *Enterobacteriaceae* as a family has been seen to fluctuate in the presence and absence of organic and chemical fertilizers. Its prominence on non-netted melons, a smooth surface, should be further explored to determine if environmental factors like organic fertilizers such as manure being introduced in conjunction with climatic changes (humidity or UV exposure) allow for an evolutionary advantage over other bacteria. Where organic fertilizers upregulate the presence of this family both in the soil and the surface of produce, chemical sprays such as soluble nitrogen for plant uptake can downregulate the relative abundance of *Enterobacteriaceae* [41–45]. In addition to that, Arizona and North Carolina did not have *Enterobacteriaceae* as a core in netted melons, possibly due to these two region's general environmental differences to the other regions in the study. Arizona is known for hot, dry climates and sandy loamy soil packed with minerals [46], which could impact the ability of certain bacteria like *Enterobacteriaceae* to colonize and/or survive on the melon surface. North Carolina is the opposite of this, where there is a temperate climate in conjunction with a coastline with consistent rainfall and a range of organic soils that are well-drained [47, 48]. Differences in environmental climate and weather can be applied to the other regions as well and just what factors play a role in composition among the regions. Consistent rainfalls in regions like Georgia and North Carolina could impact the prevalence of microorganisms on the surfaces of produce when including surrounding environments like ponds or irrigation canals for colonization of microorganisms [49, 50]. Diving deeper into the core family found, the relative abundances of *Bacillaceae* could play an important role as a biocontrol against plant diseases like *Fusarium* wilt as well as pest control [36, 51]. This disease is more prominent in wetter regions because of the lack of soil

drainage and relatively higher temperatures to reduce water retention. Lastly, *Pseudomonadaceae* has been studied for its role in the deterioration of vegetables and fruits [21], with some studies linking species of *Pseudomonadaceae* along with other microbes to protagonistic growth for fruits and vegetables [52, 53]. Utilizing these families in appropriate combinations could have implications in the future for plant health and finding better approaches for overall crop yield.

Contributions to bacterial community clustering for the two major commercial growing regions, Arizona, and California, were different over the four-year period that netted melons were collected from these locations. Bacterial composition shifts over the four years in Arizona were driven mostly by turnover of the bacteria, whereas, melons grown in California, the shifts were driven by nesting of bacteria each year. This meant that the bacteria found on netted melons for Arizona were turning over and had different bacteria being present on the melons over the years they were harvested. In California, netted melons had nested bacteria, meaning that a group of the bacteria present were consistent over the years of sampling. Potentially the main factors that divide the bacterial composition for these regions are within their geographical microclimates like the average temperatures, rainfall, and wind. Possibly due to the hot and dry climate of Arizona, the bacteria have not developed strategies resistant to desiccation and cell death under harsher temperatures [49, 54, 55]. California is the opposite, where the bacterial families continued to be present every year that the melons were harvested rather than being replaced, which could be due to California's wetter or more humid climates with relatively more fertile soil conditions, that are a little more acidic, and void of too many minerals [47]. These favorable conditions in California, could allow for bacteria to thrive, without a need for developing mechanisms for continual colonization on the surfaces of netted melons compared to those in Arizona.

## Conclusion

This study looked at the bacterial diversity and composition of different melon varieties grown across multiple regions in the United States. It is one of the first studies to look at variation across multiple varieties, seasons, as well as several growing regions to understand variation in multiple regions for commercial melon production. We concluded that variation in bacterial composition is primarily accounted for regionality that the melons are grown alongside seasonal variation rather than the reticulation or netting surfaces of the melons. However, the type of melon netting does still have an impact on the bacterial diversity and composition of the melon rind. This study also found temporal shifts on bacterial composition among the two major commercial growing regions that further supported the region contributing to differences on the surfaces of melons, but the reasons driving these shifts were different between the two regions. In conjunction with previous studies that looked at surface and stem composition of melons grown in one location, this study lays the foundation for understanding the bacterial variation and composition on the surfaces of melons. With this foundation, further studies in determining specific bacteria that are present based on the region could help the food chain and melon industry devise measures for post-harvest processing and storage to improve melon safety and shelf-life.

## Supporting information

**S1 Fig. Geographical map of sampling locations for the study.** Map of the United States with location of each field that melons were grown and harvested during this study. Location marking of the field is based on the longitude and latitude of the field. Map was generated in R using the ggplot package version 3.4.1.
(TIF)

**S2 Fig. Overall taxonomic profiling and bacterial community diversity of netted and non-netted melons.** (A)Taxonomic relative abundance of netted and non-netted melons identified at the Family level. (B) Bray-Curtis PCoA plot clustered by the year and colored by the rind netting.
(TIF)

**S3 Fig. Beta diversity of all regions collected every year for netted and non-netted melons.** (A) Bray-Curtis PCoA plot clustered by the regions in 2018. (B) Bray-Curtis PCoA plot clustered by the regions in 2019. (C) Bray-Curtis PCoA plot clustered by the regions in 2020. (D) Bray-Curtis PCoA plot clustered by the regions in 2021.
(TIF)

**S4 Fig. Taxonomic profiling of Arizona and California netted and non-netted melons.** (A) Taxonomic relative abundance looking at Arizona and California for netted melons. (B) Taxonomic relative abundance looking at Arizona and California for non-netted melons. (C) Lefser analysis for Arizona and California netted melons. (D) Lefser analysis for Arizona and California non-netted melons (LDA = 4).
(TIF)

**S5 Fig. Role of temporal distance (years the melons were collected) against bacterial variance for netted melons among Arizona and California locations.** (A) β-Sorensen dissimilarity plotted against Euclidean temporal distance using mantel correlation to assess taxa presence or absence of bacterial communities for California netted melons. (B) β-Sorensen dissimilarity plotted against Euclidean temporal distance using mantel correlation to assess taxa presence or absence of bacterial communities for Arizona netted melons. (C) β-Simpson dissimilarity plotted against Euclidean temporal distance using mantel correlation to assess taxa replacement or turnover of bacterial communities for California netted melons. (D). β-Simpson dissimilarity plotted against Euclidean temporal distance using mantel correlation to assess taxa replacement or turnover of bacterial communities for Arizona netted melons. (E) β-SNE dissimilarity plotted against Euclidean temporal distance using mantel correlation to assess taxa nestedness of bacterial communities for California netted melons. (F) β-SNE dissimilarity plotted against Euclidean temporal distance using mantel correlation to assess taxa nestedness of bacterial communities for Arizona netted melons.
(TIF)

**S1 Table. Core bacterial families of netted and non-netted melons.**
(PDF)

**S2 Table. Core bacterial families of netted melons.**
(PDF)

**S3 Table. Core bacterial families of non-netted melons.**
(PDF)

**S4 Table. Core bacterial families of Arizona and California melons.**
(PDF)

## Acknowledgments

The authors thank Greg T. Chism and Trevor Hoshiwara for assistance in statistical analysis and visualization of the data for this study.

## Author Contributions

**Conceptualization:** Madison Goforth, Victoria Obergh, Richard Park, Kerry K. Cooper.

**Formal analysis:** Madison Goforth, Craig T. Parker.

**Funding acquisition:** Bhimanagouda S. Patil, Kerry K. Cooper.

**Methodology:** Madison Goforth, Victoria Obergh, Richard Park, Martin Porchas, Kevin M. Crosby, John L. Jifon, Sadhana Ravishankar, Paul Brierley, Daniel L. Leskovar, Thomas A. Turini, Jonathan Schultheis, Timothy Coolong, Rhonda Miller, Hisashi Koiwa, Bhimanagouda S. Patil, Margarethe A. Cooper, Steven Huynh, Craig T. Parker, Wenjing Guan.

**Project administration:** Kerry K. Cooper.

**Resources:** Kerry K. Cooper.

**Supervision:** Kerry K. Cooper.

**Writing – original draft:** Madison Goforth, Kerry K. Cooper.

**Writing – review & editing:** Madison Goforth, Victoria Obergh, Richard Park, Martin Porchas, Kevin M. Crosby, John L. Jifon, Sadhana Ravishankar, Paul Brierley, Daniel L. Leskovar, Thomas A. Turini, Jonathan Schultheis, Timothy Coolong, Rhonda Miller, Hisashi Koiwa, Bhimanagouda S. Patil, Margarethe A. Cooper, Steven Huynh, Craig T. Parker, Wenjing Guan, Kerry K. Cooper.

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
