## [Decision Letter · Decision Letter 0]

18 Dec 2023

PONE-D-23-34406Bacterial diversity and composition on the rinds of specific melon cultivars and hybrids from across different growing regions in the United StatesPLOS ONE

Dear Dr. Cooper,

Thank you for submitting your manuscript to PLOS ONE. After careful consideration, we feel that it has merit but does not fully meet PLOS ONE’s publication criteria as it currently stands. Therefore, we invite you to submit a revised version of the manuscript that addresses the points raised during the review process. The reviewer #1 suggested some points to improved this manuscript. Handling editor completely agrees those comments and ask you to revise accordingly. 

We look forward to receiving your revised manuscript.

Kind regards,

Hiroshi Ezura

Academic Editor

PLOS ONE

Journal Requirements:

"{This study was supported by the USDA-NIFA-SCRI # 2017-51181-26834 through the National Center of Excellence for Melon at the Vegetable and Fruit Improvement Center of Texas A&M University. Technology and Research Initiative Fund (TRIF) provided to Kerry Cooper by the University of Arizona. No funding agency had any role in the study design, data collection and analysis, decision to publish, or preparation of the manuscript."

"This study was supported by the USDA-NIFA-SCRI # 2017-51181-26834 through the National Center of Excellence for Melon at the Vegetable and Fruit Improvement Center of Texas A&M University. Technology and Research Initiative Fund (TRIF) provided to Kerry Cooper by the University of Arizona. No funding agency had any role in the study design, data collection and analysis, decision to publish, or preparation of the manuscript."

Reviewers' comments:

Reviewer's Responses to Questions

**Comments to the Author**

1. Is the manuscript technically sound, and do the data support the conclusions?

Reviewer #1: Partly

2. Has the statistical analysis been performed appropriately and rigorously? 

Reviewer #1: Yes

3. Have the authors made all data underlying the findings in their manuscript fully available?

Reviewer #1: Yes

4. Is the manuscript presented in an intelligible fashion and written in standard English?

Reviewer #1: Yes

5. Review Comments to the Author

Reviewer #1: In the authors' study, they conducted massive 16S rRNA sequencing to investigate the bacterial diversity and composition on the rinds of melon cultivars. Their dataset covers several geographical places of US and also not limited to specific melon cultivars. Although this revewer basically agrees with publication in PLoS One, there are several concerns as described below. Most are relating the way of presentation.

1. In Fig.1, the authors compare bacterial diversity between netted and non-netted melons. But, I was really wondering whether these melons were grown under the same growth conditions. Because bacterial diversity can be fluctuated dependent the growth conditions, they should be grown in parallel on the same field at the same timing to compare bacterial diversities in a strict sense. Without such detailed information, readers cannot evaluate the authors' results.

2. Relating to 1), if growth conditions were not same in a strict sense between netted and non-netted melons, it is recommended to avoid comparing them in a single figure. Eeven if the location of production was same, they may not be comparable. Instead, it seems better to parepare different figure images for netted and non-netted melons separately.

3. Throughout the manuscript, similar figure images are repeatedly presented in different Figs. For example, Fig.1C and Fig.2B originate from completely same dataset although coloring patterns are different. Such duplication should be avoided. In addition, please consider to make manuscript more compact to make the authorss' point clear. It is not necessary to mention all things.

4. Addition of geographical maps to some figure images may be helpful for readers' understanding.

5. Almost all figure images look like collappsing. For example, it is hard to know the borders between plots and charts in each image. Also, font sizes are not unified. Please revise all figure images.

6. PLOS authors have the option to publish the peer review history of their article (what does this mean?). If published, this will include your full peer review and any attached files.

Reviewer #1: No

---

## [Author Response · Author response to Decision Letter 0]

5 Jan 2024

We thank the reviewer for their kind comments and excellent insights into improving our manuscript prior to publication. We have addressed all the comments in the revised manuscript and the changes for each specific comment are highlighted below.

1. In Fig.1, the authors compare bacterial diversity between netted and non-netted melons. But, I was really wondering whether these melons were grown under the same growth conditions. Because bacterial diversity can be fluctuated dependent the growth conditions, they should be grown in parallel on the same field at the same timing to compare bacterial diversities in a strict sense. Without such detailed information, readers cannot evaluate the authors' results.

We agree with the reviewer that it would not be appropriate to compare the netted versus non-netted melons if they were no grown in the same field under identical conditions. All melons (netted and non-netted melons) were grown in the same field under identical conditions throughout the study, all the seeds were planted at the same time in parallel to each other in the same field and harvested at the exact same time. We have added this to the Methods section of the manuscript (Lines 167 – 169) to make the point clear to readers for evaluation of the results.

2. Relating to 1), if growth conditions were not same in a strict sense between netted and non-netted melons, it is recommended to avoid comparing them in a single figure. Even if the location of production was same, they may not be comparable. Instead, it seems better to prepare different figure images for netted and non-netted melons separately.

As previously mentioned above, all melons were grown in parallel under identical conditions. We have to respectively disagree with the reviewer that comparing the netted and non-netted melons separately would be the best way to address the data. A major aspect of the study was to determine the bacterial diversity and composition of different types of melons grown under identical conditions, particularly to understand the role the netting of the melon plays in the composition of the rind microbiome. This is critical information from multiple perspectives for the melon industry including a food safety, post-harvest processing, and storage. Additionally, we feel readers can still make individual comparisons of only netted or only non-netted melons from the different locations from the figures as they are presented, while also getting the comparison between the different rind netting types.

3. Throughout the manuscript, similar figure images are repeatedly presented in different Figs. For example, Fig.1C and Fig.2B originate from completely same dataset although coloring patterns are different. Such duplication should be avoided. In addition, please consider to make manuscript more compact to make the authors' point clear. It is not necessary to mention all things.

We understand the reviewers concern about the different color patterns between Figure 1C and 2B, although they were comparing different aspects thus the different colors (Figure 1C – rind netting; Figure 2B – location) we have adjusted the years for both figures to have symbol differences and for Figure 1C to have a similar color pattern as Figure 2B (although there are less colors because of the comparison). We have also moved the original Figure 1C to the supplemental results to make the manuscript more compact. Additionally, we have moved several other figures or parts of figures to the supplemental results for the manuscript to help make the results of the study clearer. Although the number of figures has not changed due to breaking figures up to clarity (as mentioned below to address point 5 of the reviewer) those figures remaining figures have significantly less subsection figures to overall make the manuscript more compact as recommended by the reviewer.

4. Addition of geographical maps to some figure images may be helpful for readers' understanding.

We agree with the reviewer’s comment about a geographical map to help readers (particularly international readers) understand the different growing locations for the study. We have generated a map of the United States in R with the locations marked according to the longitude and latitude of the growing fields and included it in the supplemental files. The methods including R package used to generate the geographical map has also been added to the Methods section of the manuscript.

5. Almost all figure images look like collapsing. For example, it is hard to know the borders between plots and charts in each image. Also, font sizes are not unified. Please revise all figure images.

We understand the reviewer’s point about the figures in the manuscript, and figures have been adjusted, split apart, and otherwise modified to make the plots and charts and legends clearer to the reader. Additionally, we have adjusted all the figures to make sure the font sizes are uniform throughout all the figures.

---

## [Decision Letter · Decision Letter 1]

5 Feb 2024

Bacterial diversity and composition on the rinds of specific melon cultivars and hybrids from across different growing regions in the United States

PONE-D-23-34406R1

Dear Dr. Cooper,

We’re pleased to inform you that your manuscript has been judged scientifically suitable for publication and will be formally accepted for publication once it meets all outstanding technical requirements.

Kind regards,

Hiroshi Ezura

Academic Editor

PLOS ONE

Additional Editor Comments (optional):

Reviewers' comments:

Reviewer's Responses to Questions

**Comments to the Author**

1. If the authors have adequately addressed your comments raised in a previous round of review and you feel that this manuscript is now acceptable for publication, you may indicate that here to bypass the “Comments to the Author” section, enter your conflict of interest statement in the “Confidential to Editor” section, and submit your "Accept" recommendation.

Reviewer #1: All comments have been addressed

2. Is the manuscript technically sound, and do the data support the conclusions?

Reviewer #1: Yes

3. Has the statistical analysis been performed appropriately and rigorously? 

Reviewer #1: Yes

4. Have the authors made all data underlying the findings in their manuscript fully available?

Reviewer #1: Yes

5. Is the manuscript presented in an intelligible fashion and written in standard English?

Reviewer #1: Yes

6. Review Comments to the Author

Reviewer #1: I think the manuscript entitled "Bacterial diversity and composition on the rinds of specific melon cultivars and hybrids

from across different growing regions in the United States" and the figures presented in it has been revised enough. I have no further comment on this manuscript.

7. PLOS authors have the option to publish the peer review history of their article (what does this mean?). If published, this will include your full peer review and any attached files.

Reviewer #1: No

---

## [Editor Report · Acceptance letter]

27 Mar 2024

PONE-D-23-34406R1 

PLOS ONE

Dear Dr. Cooper, 

I'm pleased to inform you that your manuscript has been deemed suitable for publication in PLOS ONE. Congratulations! Your manuscript is now being handed over to our production team.

Kind regards, 

on behalf of

Prof. Hiroshi Ezura 

Academic Editor

PLOS ONE